# Electrochemical (Bio)Sensing Devices for Human-Microbiome-Related Biomarkers

**DOI:** 10.3390/s23020837

**Published:** 2023-01-11

**Authors:** Esther Sánchez-Tirado, Lourdes Agüí, Araceli González-Cortés, Susana Campuzano, Paloma Yáñez-Sedeño, José Manuel Pingarrón

**Affiliations:** Department of Analytical Chemistry, Faculty of Chemistry, Universidad Complutense of Madrid, 28040 Madrid, Spain

**Keywords:** electrochemical biosensors, electrochemical sensors, microbiome, biomarkers

## Abstract

The study of the human microbiome is a multidisciplinary area ranging from the field of technology to that of personalized medicine. The possibility of using microbiota biomarkers to improve the diagnosis and monitoring of diseases (e.g., cancer), health conditions (e.g., obesity) or relevant processes (e.g., aging) has raised great expectations, also in the field of bioelectroanalytical chemistry. The well-known advantages of electrochemical biosensors—high sensitivity, fast response, and the possibility of miniaturization, together with the potential for new nanomaterials to improve their design and performance—position them as unique tools to provide a better understanding of the entities of the human microbiome and raise the prospect of huge and important developments in the coming years. This review article compiles recent applications of electrochemical (bio)sensors for monitoring microbial metabolites and disease biomarkers related to different types of human microbiome, with a special focus on the gastrointestinal microbiome. Examples of electrochemical devices applied to real samples are critically discussed, as well as challenges to be faced and where future developments are expected to go.

## 1. Introduction

In recent decades, research in the field of the microbiome has evolved rapidly, it currently being a topic of great interest at both the scientific and social levels [1]. The human microbiome has been defined as “an ecological community of commensal, symbiotic and pathogenic microorganisms that literally share our body space and have been all but ignored as determinants of health and disease” [2]. For instance, extensive investigations on the gut microbiome have demonstrated the association of numerous diseases and conditions (gastrointestinal or otherwise) with altered levels of intestinal microbes and related biomarkers. The results of these studies are considered an essential tool of personalized medicine, offering interesting solutions to a variety of autoimmune and metabolic illnesses and improving diagnostic and treatment strategies [3]. In a very recent review article, Aggarwal et al. [4] describe the current understanding of the relationship between the microbiome and disease as well as the therapeutic effects of microbiome modulation on the host. Microbial cells are mostly located in the gut (around 60%), with a prominent role in the microbiome gut–brain axis and the gastrointestinal tract, and to a lesser extent in the oral and nasal cavities, skin, and genital surfaces (Figure 1). Alterations in the microbiome are related to various diseases, including autoimmune and degenerative processes, as well as clinical situations and conditions. Some examples are colorectal cancer (CRC), celiac disease, cirrhosis, inflammatory bowel disease (IBD), anxiety and depression, obesity, allergies, and ageing.

A thorough understanding of the human microbiome requires identifying and monitoring of small molecules and proteins produced by microorganisms and surrounding the microbiome environment that are responsible for catalytic functions and metabolic activity. These biomolecules govern interactions with host cells and are closely related to different diseases. The microbial composition is characterized by genomic and transcriptomic sequencing, while metabolomics and other “omics” identify protein pools that can be subsequently detected and quantified using (bio)sensing tools. For example, the intestinal microbiome of insulin-resistant individuals is known to have a high capacity for the synthesis of branched-chain amino acids that are detected at high concentrations in the serum of patients [5]. Obviously, the difficulty for the diagnosis lies in obtaining the necessary “omic” information on the species to be determined, and to have both sensitive and accurate analytical methods, as well as adequate samples. With this in mind, research can be limited to simply determining the presence or the absence of specific species within or outside the microbiome locus, or to establishing the relationship of the various species to each other or to the host cells.

In the overall effort for a better understanding of the human microbiome, sensors and biosensors play an important role, since their use can help the collection of a large number and complex variety of microbiome data for the assessment of the presence and evolution of a specific disease. However, due to the wide variety of diseases associated with microbial activity and diversity, and the effect of other active biomolecules not necessarily derived from microorganisms, (bio)sensing tools with the best analytical characteristics of sensitivity, selectivity, reproducibility, accuracy and multiplexing ability are needed. Indeed, related to multiplexing, better diagnosis requires multiple detection with complex clinical samples and with the ability to seek differences in the composition of a given location in the presence or absence of the disease.

The possibility of using microbiota biomarkers to advance disease monitoring has raised great expectations, also in the field of bioelectroanalytical chemistry, which has demonstrated the ability to develop point-of-care (PoCs) detection tools for a variety of biomolecules. The inherent advantages of electrochemical transduction, such as high sensitivity, rapidity of response, and possibility of miniaturization, together with the potential of new nanomaterials to improve both the design of electrode platforms and the performance of the resulting electrochemical biosensors, make probable a huge development in the coming years, providing a better understanding of human microbiome entities.

The potential of biosensors to bring the study of the microbiome into the realms of clinical diagnostics and mass data collection was reviewed [3]. More recently, Yadav et al. [6] reviewed advances in electrochemical sensors involving nanomaterials for clinically associated human and gut metabolites, including those from the microbiome. Merkoçi’s team also discussed opportunities and challenges in developing new nanotechnology-based diagnostic devices for microbiome research [7].

However, to our knowledge, there is no review article in which the opportunities and potential for electrochemical biosensors to advance knowledge and detection of the human microbiome are critically discussed. That is why this review article aims to provide an overview of recent applications of electrochemical (bio)sensors for the monitoring of microbial metabolites and the detection of disease biomarkers related to different types of human microbiome, with a special focus on the gastrointestinal microbiome. To illustrate the critical discussion, examples of electrochemical devices applied to real samples are considered.

## 2. Gastrointestinal Microbiome

The study of the gastrointestinal microbiome is currently an exciting area for managing the health of the whole organism. The reason is because it modulates several physiological functions, including the immune system [8] and, through the microbiome gut–brain axis, also behaviour and mental state, causing or preventing anxiety and depression [9]. The human gut is a dynamic environment, in which microorganisms constantly interact with the host via their metabolic products. The decisive role in both processing signals from the environment and distributing them to the organism has led to the gastrointestinal microbiome being called the “fifth organ” [8]. The most recent studies on the involvement of gut microbiota in the pathogenesis of many diseases as well as the different strategies used to manipulate the gut microbiota in the prevention and treatment of disorders have been reviewed [10]. The outcomes of this complex relationship are the main pathways used to modulate the functionality of organs and systems such as the brain and the immune system, which are involved in host health and disease, and also encompass a range of activities that extend to nutrient or drug metabolism and the immune response to pathogens [11]. The gastrointestinal microbiome modulates several physiological functions such as digestion, energy metabolism, immune system development, and infection prevention [12,13,14].

Biomarkers of gastrointestinal functionality in animal nutrition and health were reviewed by Pietro et al. [15]. Metabolites produced by gut microbiota, such as trimethylamine N-oxide (TMAO), trimethylamine (TMA), short-chain fatty acids (SCFAs), and indole derivatives, contribute to various human diseases [6], such as metabolic and cardiovascular diseases [16,17], cancer and inflammation [18], depression [19], and colorectal cancer (CRC) [20].

Microbiome-derived metabolites, and in particular polyamines, are known to be involved in carcinogenesis in both animal models and humans [21]. Gut microbiota degrades nutrients rich in trimethylamine (TMA)-containing substances, such as choline, carnitine, and lecithin, to produce TMAO, which has been associated with an increased risk of developing complex illnesses such as cardiovascular diseases (CVDs), CRC [22,23], chronic kidney diseases [24], and diabetes. As a result, TMAO is considered a critical prognostic and diagnostic biomarker and its analytical monitoring is critical in health management.

As can be seen in Appendix A, which summarizes the analytical characteristics of selected electrochemical (bio)sensors for biomarkers of the human microbiome and related biomolecules applied to clinical samples, several electrochemical (bio)sensors have been reported for the determination of TMAO. A molecularly imprinted polymer (MIP) prepared from polypyrrole (PPy) on hydrolyzed indium tin oxide (ITO)-coated glasses was reported by Lakshmi et al. [25]. The MIP was made using a chemical oxidation polymerization technique in the presence and absence of TMAO, and the detection response was provided by the peak current reduction of the target recorded by differential pulse voltammetry (DPV). The method exhibited a good sensitivity, with a LOD value of 1 μg mL^−1^ and a relatively narrow linear dynamic range between 1 and 15 μg mL^−1^, although it is claimed to be appropriate for the analysis of biological fluids. The sensor was applied to the analysis of spiked urine with recovery percentages of around 100%.

An enzyme electrochemical biosensor for TMAO was reported using an active chimeric variant of TMAO reductase in combination with formate dehydrogenase (TorA-FDH) immobilized on a glassy carbon electrode and coated with a dialysis membrane. The biosensor operated at an applied potential of −0.8 V vs. Ag/AgCl under ambient air conditions in the presence of methyl viologen as the redox mediator. A measuring range of 2–110 μM with a detection limit of 2.96 nmol L^−1^ TMAO was obtained, and the biosensor was applied to spiked serum samples [26]. More recently, the same group reported another enzyme biosensor for TMAO involving three enzymes: TMAO reductase (TorA), glucose oxidase (GOD) and catalase (Cat). An oxygen anti-interference membrane composed of GOD, Cat and polyvinyl alcohol (PVA) hydrogel was prepared, and the TMAO biosensor was constructed by purifying *Escherichia coli* TorA under anaerobic conditions and immobilizing it on the surface of a carbon electrode, which was subsequently coated with the O_2_ scavenging membrane. The detection potential was the same as that in the previous work, and methylviologen was also employed as the redox mediator. The sensor signal was linearly dependent on TMAO concentrations between 2 µM and 15 mM. Measurements of TMAO concentration were performed in 10% human serum, where the lowest detectable concentration was 10 µM TMAO [27]. Yi et al. [28] developed a unique TMAO detection technique based on microbial electrochemical reduction of the biomarker with *Shewanella loihica* PV4, which used TMAO as an electron acceptor for aerobic respiration. Direct attachment of the cells onto carbon cloth electrodes allowed the measurement of cathodic currents proportional to TMAO concentrations over a wide linear range up to 250 μM, with low LOD (5.96 μM) and high sensitivity (23.92 μA mM^−1^). The developed method allowed the determination to be completed in 600 s, providing an accuracy rate of 90% in serum.

Small molecules produced by microbial fermentation of carbohydrates and proteins are related to insulin resistance, obesity, and immune diseases. Among these metabolites, SCFAs and their anions and amino acid derivatives (e.g., indol) are of great interest, as they act as signaling molecules in the host–microbiota interaction [29]. For instance, butyrate and propionate anions of SCFA in feces show modification in their levels depending on the clinical state (acute or in remission phases) of patients suffering from rheumatoid arthritis (RA) [30]. Moreover, acetic, butyric, propionic, valeric, isobutyric, and isovaleric acids, and their respective anions, are the main fermentation end-products of non-digestible carbohydrates that serve as energy sources for gut epithelial cells. They modulate cytokine production and induce expansion of regulatory T cells [31] and have been associated with kidney diseases, hypertension, and inflammation [32]. Numerous efforts have been focused on providing evidence for the role of SCFAs in the relationship between the intestinal microbiome and host health [33]. Among others, the important implication of SCFAs in the development of inflammatory bowel disease (IBD) has been demonstrated. IBD is a term involving two conditions: Crohn´s disease and ulcerative colitis, both characterized by chronic inflammation of the gastrointestinal tract, whose accurate detection and diagnosis require extensive sample preparation and expensive equipment. Therefore, the development of specific (bio)sensors for SCFAs and related molecules is highly desired. However, so far, the number of methods with sufficient sensitivity and selectivity to tackle this task is small, and the few reported strategies have not been practically applied to biological samples. For example, enzymatically modified microfabricated platinum electrodes were used for the amperometric quantification of acetate and propionate, mediated by the oxidation of hydrogen peroxide [34]. As Figure 2 shows, two different enzyme systems were used: the amperometric detection of acetate was enabled by a combination of acetate kinase (AK), pyruvate kinase (PK) and pyruvate oxidase, whereas propionate CoA-transferase (PCT) and short-chain acyl-CoA oxidase (SCAOx) were used as the catalytic strategy for propionate quantification. The developed methods provided similar analytical characteristics, with linear ranges ranging up to 1.4 or 1.5 mM for acetate and propionate, respectively.

An impedimetric sensor for the real-time detection of gut-microbiota-generated SCFAs was reported by Yavarinasab et al. [35]. Figure 3 displays the electrochemical platform consisting of interdigitated gold electrodes modified with ZnO and polyvinyl alcohol (PVA). EIS measurements of the acids were performed in the liquid phase at room temperature for in vitro detection of acetic acid, propionic acid, and butyric acid, which account for more than 95% of SCFAs in the intestine at concentrations ranging from 0.5 to 10 mg mL^−1^. The sensor detected the level of SCFAs in bacterial isolates (*L. plantarum* and *E. coli*) and identified them with high accuracy using only 2.5 μL of sample. Other types of SCFA biosensors based on living microorganisms have also been reported, with an interesting approach of possible quantification using microbial fuel cells reported by Kaur et al. [36]. The proposed biosensor array was able to measure individual acetate, propionate, and butyrate concentrations down to 5 mg L^−1^ and up to 40 mg L^−1^. However, the detection range was rather limited for real applications in clinical samples.

Succinate is a microbiota-derived metabolite with a key role in governing intestinal homeostasis and related to a microbiome signature [37]. It is involved in several metabolic pathways, with enhanced levels derived from gut microbiome dysbiosis and increasing intestinal permeability [38]. Succinate concentration is clearly elevated in inflammatory-related health conditions, including obesity and type 2 diabetes (T2D) [39,40], and it was validated as a surrogate biomarker of poor metabolic control in patients suffering these illnesses [41]. High circulating levels of succinate in human obesity are linked to specific gut microbiota because of an abundance of succinate-producing microorganisms and scarce succinate-consuming microbes. Furthermore, alterations in circulating succinate are also related to carbohydrate metabolism and energy production [38]. The role of succinate in the regulation of intestinal inflammation has been reviewed by Connors et al. [42]. No electrochemical sensors for the determination of succinate have been found in the reviewed literature. However, the electrochemical oxidation of succinic acid in aqueous solutions using boron-doped diamond electrodes was investigated [43], although the absence of analytically useful responses precludes the preparation of a suitable sensor. Electrochemical biosensors for the detection of this species have not been developed either.

Monitoring microbiome-related biomarkers of gastrointestinal inflammation is crucial to provide relevant information on the interactions of the gastrointestinal tract with the environment and to know to what extent the functionality of the gastrointestinal barrier is maintained. However, data on the detection of gut biomarkers for these disorders are scarce. There is wide evidence that cytokines play a crucial role in the pathogenesis of IBD [44]. Relevant targets are pro-inflammatory cytokines such as IL-6, IL-12, IL-23, and IL-21, as well as anti-inflammatory cytokines, such as IL-10 and TGF-β. Pro-inflammatory interleukins and tumor growth factor (TNF-α) are known to contribute to increased intestinal permeability leading to translocation of bacteria and toxins and ultimately inflammation.

Reflecting this key role of cytokines in relation to the intestinal microbiome and derived inflammatory diseases, a large number of electrochemical (bio)sensors have been reported in recent years for the single or multiple determination of interleukins, chemokines and other cytokines in clinical samples [45,46]. However, methods devoted to interleukins directly related to the gut microbiome and IBD disorder are very scarce. The incorporation of nanomaterials in the construction of electrochemical biosensors allows the detection of ILs with high sensitivity and wide dynamic ranges in comparison with other techniques, as well as the possibility of application to body fluids others than serum, such as saliva or sweat, where these biomarkers are present in much lower concentration [47]. A general review of recent advances and possibilities with electrochemical biosensors for cytokine profiling has been published [48]. In addition, recent progress in nanomaterial-based electrochemical biosensors for the detection of interleukins has also been reviewed [49]. A recent example of electrochemical biosensors for ILs related to IBD is the impedimetric immunosensor reported by Frias et al. [50] for IL-10, an interleukin secreted in patients at the early stage of inflammation. The biosensor involves the fabrication of a microfluidic lab-on-chip device using graphene-foam flexible electrodes functionalized with pyrene carboxylic acid by π complexation, and the covalent immobilization of the anti-IL-10 antibodies. EIS measurements allowed IL-10 quantification in artificial saliva in the range from 10 to 100 fg mL^−1^ with a LOD value of 7.89 fg mL^−1^.

Along with cytokines, C-reactive protein (CRP) is also a recommended biomarker for early detection of IBD [51]. A multiplexed sensor for the continuous monitoring of IL-1β and CRP in human eccrine sweat was reported by Jagannath et al. [52]. A replaceable sweat-sensing strip functionalized for the specific targets and mounted on a wearable transducer consisting of a screen-printed two-electrode system was prepared. Impedimetric measurements were carried out upon immobilization of the respective capture antibodies using the cross-linker DTSSP (3,3′-dithiobis (sulfosuccinimidyl propionate). Dynamic ranges from 0.2 to 200 pg mL^−1^ IL-1β and up to 10 ng mL^−1^ CRP with LOD values of 0.2 pg mL^−1^ and 1 pg mL^−1^, respectively, were attained. The sensor was applied to the determination of both biomarkers in spiked sweat samples collected from healthy individuals and to continuous on-body IL-1β measurements. Appendix A summarizes the analytical characteristics claimed for other electrochemical biosensors for CRP [53,54].

Myeloperoxidase (MPO) is a specific marker of neutrophil activity [55]. The number of neutrophils in mammals has been positively correlated with tissue MPO levels, in turn related to intestinal inflammation [56] and intestinal permeability [57]. In humans, fecal MPO levels have been related to IBD disease activity [58]. A strategy for the electrochemical detection of MPO, with a microfluidic device, involved the use of streptavidin-functionalized magnetic microbeads (Strep-MBs) and biotinylated antibodies (Figure 4) [59]. Quantification of the biomarker through the measurement of its peroxidase activity allowed a LOD of 0.004 ng mL^−1^ MPO to be obtained. The developed method was successfully applied to human plasma from healthy individuals and patients with coronary ischemia.

More recently, a trimetallic CuPdPt nanowire network dropped onto a glassy carbon electrode was proposed as an electrochemical platform for immobilization of anti-MPO antibodies and the amperometric detection of MPO activity through the measurement of the H_2_O_2_ reduction current, which was proportional to the MPO concentration over the 100 fg mL^−1^ to 50 ng mL^−1^ range. The high sensitivity made it possible to apply the method to the analysis of human serum with good results [60]. Additionally, an electrochemical approach to measure MPO based on an immunoassay scheme involving immobilization of MPO-capture antibody on a polystyrene dipstick was implemented. After immobilization of the target, the ability of MPO to participate in enzymatic pseudohalogenation and catalase-like reactions involving, respectively, MPO/SCN^−^/H_2_O_2_ and MPO/H_2_O_2_, was harnessed, and amperometric detection was performed by monitoring the response of H_2_O_2_ at −0.2 V vs. Ag/AgCl with a nitrogen-doped carbon-nanotube (N-CNTs) electrode. The method allowed detection of 60 μg L^−1^, which was suitable for the detection of MPO in human saliva [61].

Related to MPO, other intestinal enzymes such as diamine oxidase (DAO) and intestinal alkaline phosphatase (iALP) have great interest. On the one hand, mucosal damage in the small intestine and enhancement of intestinal permeability inversely correlates with DAO activity as a catalyst of diamine oxidation. This is particularly relevant in humans affected by Crohn’s disease, where mucosal DAO activity is nearly 50% lower than in healthy counterparts [62]. Moreover, this enzyme has an essential role in the degradation of exogenous histamine in the intestine, a good correlation existing between histamine intolerance and low concentration and/or activity of DAO [63]. Furthermore, iALP is involved in several physiological roles in the gastrointestinal tract, such as dephosphorylation of bacterial lipopolysaccharides (LPS) and regulation of lipid absorption, thus playing a protective role against both LPS-induced inflammation and Type 2 diabetes in humans [64]. No reports of electrochemical (bio)sensors for DAO have been found in the revised literature. However, due to its more widespread interest, there are numerous reported amperometric, impedimetric, and potentiometric biosensors applied to the determination of alkaline phosphatase in different types of samples. An overview of the application of electrochemical (and optical) biosensors for alkaline phosphatase in cell cultures has been published [65]. An illustrative example is an impedimetric immunosensor for serum alkaline phosphatase detection based on electrochemically engineered Au-nano-dendroids and graphene oxide nanocomposite [66]. It involved label-free impedance measurements on a modified SPCE with immobilized anti-ALP antibodies (Figure 5A) and provided a linear dynamic range between 100 and 1000 U L^−1^. The immunosensor was applied to clinical serum samples. Figure 5B shows another electrochemical biosensor that used aminoferrocene (AFC) labelled on ssDNA by conjugating it with phosphate groups as an electroactive probe for ALP activity detection. The thiolated ssDNA at 3′terminals was self-assembled on the surface of an Au electrode via S-Au bonding, and, after incubation with ALP, the removal of phosphate groups from the 5’ terminus of ssDNA was catalyzed, and the AFC probe cannot be labelled on ssDNA. This strategy provided a linear range between 20 and 100 mU mL^−1^ [67].

Among microbiome-produced metabolites, indole and its derivatives are relevant biomarkers for various types of inflammation, including that associated with ageing and central nervous system inflammation [68]. These biomolecules appear in the gastrointestinal microbiome as microbial metabolic products of tryptophan. *Lactobacillus reuteri* produces indole-3-aldehyde (I3A), whereas pathogenic *E. coli* strains secrete various indole derivatives, including I3A and indole-3-acetic (I3AA). In addition to these, other metabolites of indole structure associated with gut bacteria have been found to be involved in various non-infectious diseases [69]. It is important to point out that, due to the interest in the determination of these species in other fields, such as agrochemicals, the number of (bio)sensors reported in the literature for these derivatives is high. However, very few electrochemical biosensors have been applied to clinical samples. A recent example is the simple amperometric sensor for the determination of indole in plasma prepared with an SPCE modified with carbon nanotubes and chitosan (MWCNTs/CS/SPCE). This modified surface improved the electron transfer oxidation reaction of indole, providing a linear range by DPV of 5–100 μg L^−1^ indole and a LOD value of 0.5 μg L^−1^ [70]. The modified electrode was employed to determine plasma indole in healthy pregnant women and gestational diabetes mellitus (GDM) patients, with results (5.3 (4.1–7.0) μg L^−1^ and 7.2 (4.5–9.4) μg L^−1^, respectively) consistent with those obtained by a chromatographic method. The elevated indole levels in GDM patients suggested that indole might play a relevant role in diabetes mellitus. A voltametric sensor was developed by Moncer et al. [71] for the detection of 5-hydroxyindole-3-acetic acid (5-HIAA), a carcinoid cancer biomarker, in human serum, urine, and plasma, using a glassy carbon electrode modified with a molecularly imprinted polypyrrole. By recording the DPV current responses, a highly selective and sensitive method towards the target molecule was developed with a LOD value of 5 × 10^−12^ mol L^−1^ and a wide linear range between 5 × 10^−11^ and 5 × 10^−5^ mol L^−1^.

Among the high variety of proteins that are associated with the gastrointestinal microbiome and related diseases, those for which electrochemical (bio)sensors are available should be mentioned. For instance, intestinal fatty acids binding protein (iFABP) plays a relevant role as a biomarker of intestinal inflammation related to changes in the microbiome. Determining the levels of this protein in serum or plasma provides information about intestinal barrier dysfunction. In addition, its detection in urine or blood has been reported as a promising non-invasive method for identifying patients with acute mesenteric ischemia (AMI), provoked by an inadequate blood supply to the intestine [72]. An electrochemical biosensor for iFABP was fabricated using gold interdigitated electrodes functionalized with the specific capture antibody, sandwiching the protein with a gold-nanoparticle-modified detection antibody. A label-free impedimetric assay was implemented, providing a dynamic range encompassing the concentration of iFABP in urine below the critical concentration of 2.7 ng mL^−1^ and a LOD of 0.68 ng mL^−1^.

Other protein of interest is the one known as PAP (protein associated with pancreatitis), whose expression is proportional to the microbial response to infection and can be used as a non-invasive biomarker of the course of IBD in combination with other markers of inflammation measured in plasma such as CRP. In addition, fecal calprotectin (CALP) is a very sensitive marker for inflammation in the gastrointestinal tract, also allowing the differentiation of IBD from irritable bowel syndrome (IBS) and other diseases with different inflammatory patterns such as Crohn’s disease and ulcerative colitis [73]. A non-enzyme sandwich-like immunosensor has been reported for the determination of CALP involving immobilization of the capture antibodies on glassy carbon electrodes modified with polydopamine-decorated carbon nanotubes functionalized with gold nanoparticles (Au@MWCNTs) [74] (Figure 6). A strategy for signal amplification was employed involving the use of high-electrocatalytic PtNi nanospheres on ultrathin Cu-Fe(III) meso-tetra(4-carboxyphenyl)porphine chloride (PtNi@TCPP(Fe)) nanosheets as carrier tags for immobilization of CALP antibodies. The reduction current of H_2_O_2_ at this platform provided a calibration plot over the 200 fg mL^−1^ to 50 ng mL^−1^ antigen linear range, which is useful for the analysis of human serum.

In recent years, several studies have highlighted the role of the microbiome in the pathogenesis of autoimmune diseases [75]. Alterations in the intestinal flora and modification of the microbiome in the intestinal tract have been claimed as important factors in the pathogenesis of rheumatoid arthritis (RA) and multiple sclerosis (MS), among other diseases. There is increasing evidence that various microbial metabolites generated from carbohydrates, proteins, and bile acids profoundly regulate the immune system via host receptors and other target molecules. Importantly, microbial metabolites act bidirectionally to promote both tolerance and immunity to effectively fight infections without developing inflammatory diseases [76]. Although gastrointestinal commensal bacteria have been found to be implicated in the development of these diseases, the mechanisms underlying the relationship of human systemic autoimmunity with the microbiome have not yet been identified. Among the few biomarkers specifically derived from commensal bacteria, microbiome-associated lipopeptides are markers of neurodegeneration and related diseases such as MS [77]. For instance, a gastrointestinal- and oral-bacteria-derived lipodipeptide, Lipid 654, which functions as a Toll-like receptor 2 ligand, was found to be expressed at significantly lower levels in the serum of patients with MS than in healthy individuals [78].

The composition and status of the microbiome also have a crucial role in the initiation and progression of RA, an autoimmune disorder with increased morbidity and mortality characterized by chronic inflammation of the synovial joints leading to significant pain, swelling, and disability. Gut dysbiosis has been reported in RA and other inflammatory rheumatic diseases, including juvenile inflammatory arthritis and ankylosing spondylitis [79,80]. It has been shown that stools from patients with over-abundant *Prevotella* microorganisms relative to healthy controls were all seropositive for rheumatoid factor (RF) and anticitrullinated peptide autoantibodies (ACPAs) [81], both considered as specific biomarkers for this disease [82]. Examples of electrochemical biosensors for these biomarkers are given in the next section.

## 3. Oral Microbiome

The oral cavity has the second-largest and second-most diverse microbiota after the gut, harboring over 700 species of bacteria [83]. Therefore, the densely populated microbial communities make the oral microbiome an ideal source for the discovery of biomarkers. A proteomic analysis of saliva from healthy individuals was reported by Sivadasan et al. [84], resulting in the identification of 1256 human proteins of microbial origin. Thus, in recent years, saliva has played a central role in the diagnosis of oral and systemic diseases. There is evidence that changes in environmental conditions favor oral diseases and increase the potential for pathogenicity. Regarding this, the relationship between periodontal disease and systemic conditions including cancer, rheumatoid diseases, and diabetes mellitus has been reported in several articles [85,86,87,88].

Alteration of microenvironments in the oral cavity of normal individuals may change the microbial composition of their saliva [89], inducing pro-inflammatory responses in oral epithelial cells by activating several chemokines [90], producing short-chain organic acids and chronic inflammation caused by bacterial infection responsible for tumorigenesis [91], or resulting in the secretion of matrix metalloproteinases MMP-9 and MMP-13 (collagenase3), which contribute to oral squamous cell carcinoma (OSCC) metastasis [92] or other cancer biomarkers such as myeloid-related protein 14 (MRP14), CD59, and Mac-2-binding protein (M2BP). Several electrochemical biosensors have been reported for the detection of oral microbiome metabolites, although only a few of them have been applied to saliva. Electrochemical biosensors for the detection of matrix metalloproteinases MMPs were reviewed by Zhou [93]. An interesting example is the immunoplatform constructed for the determination of MMP-9, involving the immobilization of a capture antibody (cAb) on carboxylated magnetic microbeads (cMBs) and the implementation of a sandwich-type immunoassay using poly-HRP for signal amplification [94]. The resulting magnetically assisted immunosensor provided a linear range between 0.03 and 2 ng mL^−1^ and a LOD value of 13 pg mL^−1^MMP-9. Another MBs-based sandwich immunoassay for the amperometric determination of MMP-9 was developed where the cAb-MBs immunoconjugates were sandwiched with a biotinylated detector antibody (biotin-dAb) further labelled with a commercial streptavidin-horseradish peroxidase (Strep-HRP) polymer. The developed immunoplatform achieved a LOD value of 2.4 pg mL^−1^ MMP-9, and the method was applied to the determination of endogenous MMP-9 in both cancer cell lysates and serum samples of patients diagnosed with different subtypes of breast cancer [95].

Quantification of proteins derived from the oral microbiome in saliva makes it possible to detect autoimmune diseases that in some cases are in turn also related to the presence of cancer. For example, it is known that salivary levels of inflammatory cytokines involved in the immune response are significantly higher in RA and OSCC or tongue squamous cell carcinoma (TSCC) [96]. This is the case for the interleukins IL-8 or IL-1α and vascular endothelial growth factor A (VEGF-A) [97].

An overview of electrochemical sensors targeting salivary biomarkers was published by Mani et al. [98]. Several electrochemical biosensors have been developed for the detection of ILs [99] and specifically for IL-8, as a biomarker of oral cancer and other types of cancer. The normal concentration of IL-8 in human saliva is in the 200–300 pg mL^−1^ range, whereas patients suffering from oral cavity and oropharyngeal squamous cell carcinoma have IL-8 concentrations higher than 720 pg mL^−1^ [100,101]. This difference makes saliva, a harmless extraction sample, a very suitable medium for monitoring this type of disease. Figure 7 shows some representative examples of recent electrochemical biosensors for IL-8.

Bathia et al. prepared an immunosensor for salivary IL-8 involving a polyenzyme label based on biotinylated diaphorase and neutravidin. Figure 7A shows as the label was conjugated after covalent immobilization of anti-IL-8 on silane copolymer-modified ITO electrodes and implementation of a sandwich-type immunoassay with the antigen and a biotinylated detection antibody. The use of electrochemical–enzymatic redox cycling, with Os(bpy)_2_Cl_2_ as the electron mediator and NADH, provided high signal amplification as well as low nonspecific responses using chronocoulometry as the electrochemical technique, thus achieving a LOD value of 1 pg mL^−1^ IL-8 [101]. A label-free immunosensing approach was proposed using synthesized silver molybdate nanoparticles (β-Ag_2_MoO_4_ NPs) as the coating of ITO electrodes for covalent immobilization of anti-IL-8 (Figure 7B). The resulting immunoplatform reached a detection limit of 90 pg mL^−1^ and was applied to spiked saliva [102]. Verma et al. [103] developed an electrochemical immunosensor for non-invasive detection of oral cancer using ITO electrodes modified with gold nanopart-cle-reduced graphene oxide (AuNPs-rGO) as a platform for the label-free determination of IL-8 (Figure 7C). After immobilization of the specific capture anti-IL-8 antibody, the resulting immunosensor showed a linear dynamic range of 500 fg mL^−1^–4 ng mL^−1^, a LOD value of 72.73 ± 0.18 pg mL^−1^, and very fast detection (9 min). The immunosensor was applied to spiked human saliva. Among the various electrochemical biosensors reported by Prof. Sezgintürk’s group for the determination of IL-8 [104,105,106], Figure 7D displays a scheme of the biosensor prepared using an ITO electrode modified with 6-phosphonohexanoic acid (PHA) for the immobilization of anti-IL-8 and impedimetric detection [105]. The interest in using phosphonic acids in the preparation of electrochemical biosensors derives from their ability to spontaneously produce self-assembled monolayers (SAMs) on different electrode surfaces, including metal oxides. In addition, the SAMs of phosphonates provide a suitable matrix for immobilization of biomolecules [107]. The method developed with the IL-8-anti-IL-8-PHA-ITO immunosensor, using EIS measurements with ferro-ferricyanide as the electrochemical probe, exhibited a linear range between 0.02 pg mL^−1^ and 3 pg mL^−1^ and a low detection limit of 6 fg mL^−1^. The authors claimed satisfactory results in the analysis of real saliva and human serum.

Electrochemical biosensors for the determination of microbiome-related salivary cytokines other than IL-8 have also been reported. For example, Aydin and Sezgintürk proposed the use of 8-PHA-ITO electrodes for the preparation of an immunosensor for IL-1β applied to saliva and serum [108]. More recently, our group reported the application of electro-click methodology for the construction of a novel electrochemical immunosensor for IL-1β. The strategy involved the binding of ethynylated IgG to azide-MWCNT modified electrodes by an electrochemically synthesized Cu(I)-catalyzed cycloaddition reaction. Once the capture antibody was immobilized on IgG-MWCNTs, a sandwich-type immunosensor using biotinylated anti-IL-1β labelled with alkaline phosphatase-streptavidin (ALP-strept) as the detection antibody was implemented. DPV measurements with the 1-naphthylphosphate/1-naphthol system provided a LOD value of 5.2 pg mL^−1^, the immunosensor being applied to the determination in human saliva [109].

The ability of electrochemical biosensors to be used for multiplexed determination has been exploited for the analysis of cytokines in raw saliva. An illustrative example is the magnetically assisted bioplatform developed for the determination of IL-8 protein and its associated messenger RNA IL-8 mRNA. The strategy involved the use of carboxylated MBs, specific antibodies against IL-8, a specific hairpin DNA sequence for IL-8 mRNA, and dual screen-printed carbon electrodes (SPdCEs) [110]. Amperometric detection using the H_2_O_2_/HRP system mediated by hydroquinone (HQ) provided detection limits of 72.4 pg mL^−1^ IL-8 and 0.21 nmol L^−1^IL-8 mRNA. A dual electrochemical immunosensor was also prepared for the simultaneous determination of IL-1β and tumor necrosis factor alpha (TNF-α) in saliva and serum using SPdCEs modified with functionalized double-walled carbon nanotubes (DWCNTs) [111]. The capture antibodies were immobilized on HOOC-Phe-DWCNTs/SPdCEs by means of the polymeric coating Mix&Go™, and sandwich type immunoassays were implemented with amperometric signal amplification through the use of poly-HRP streptavidin conjugates and the H_2_O_2_/HRP/HQ enzymatic/redox system. The developed method allowed ranges of linearity extending between 0.5 and 100 pg mL^−1^ and from 1 to 200 pg mL^−1^ for IL-1β and TNF-α, respectively.

There is evidence that the oral microbiome plays a role in the etiology and progression of various autoimmune diseases, including RA and systemic lupus erythematosus (SLE) [112]. It has been demonstrated that several organisms in the oral microbiome causing periodontal infection are linked to the RA disease. The common pathogen *Porphyromonas gingivalis (P. gingivalis)* expresses a bacterial protein arginine deiminase that can citrullinate host peptides, these inducing the formation of anticitrullinated protein antibodies (ACPAs) [112]. This microorganism also carries heat shock proteins (HSPs) that may trigger auto-immune responses in subjects with RA. Among the different types of ACPAs that are used for the diagnosis of RA, anti-cyclic citrullinated peptide (anti-CCP) and anti-citrullinated enolase peptide (anti-CEP) can be detected in patients with both RA and periodontal disease [113].

The number of electrochemical biosensors devoted to ACPAs detection is relatively low. Furthermore, no publications on salivary applications in the context of RA have been found, although they do exist for human serum analysis, where the cut-off value for positivity is anti-CCP > 25 U mL^−1^. In this context, an electrochemical immunosensor was prepared using screen-printed electrodes modified with poly(aniline) (PANI) and MoS_2_. A citrulline-containing cyclic filaggrin peptide (21-mer) to explicitly recognize anti-CCP antibody was covalently attached to this surface, and a sandwich-type immunoassay was established with anti-CCP trapped in an interfacial polymerized PANI-AuNPs for signal amplification [114]. Using SWV, the achieved LOD value was 0.16 IU mL^−1^ anti-CCP.

A dual electrochemical biosensor involving carboxylated- or neutravidin-functionalized MBs and dual screen-printed carbon electrodes was developed for the simultaneous determination of anti-CCP and rheumatoid factor (RF), an autoantibody widely used as RA biomarker. Sandwich-type biosensors were constructed by involving Fc fragments of IgG Fc(IgG) and biotinylated CCP to form CCP-biotin-Neutr-MBs for the specific immobilization of RF and anti-CCP, respectively, followed by conjugation with the respective HRP-IgM and HRP-IgG. Amperometric detection using the H_2_O_2_/hydroquinone (HQ) system provided LOD values for RF and anti-CCP of 0.8 and 2.5 IU mL^−1^, respectively. The simultaneous determination can be completed in about two hours using a simple protocol and a sample volume (25 μL) four times smaller than that required by the ELISA method [115].

## 4. Nasal Microbiome

Nasal mucus and secretions constitute a first line of defense of the respiratory tract and are responsible for eliminating air pollutants and preventing microbes from entering the body [116]. At the same time, the nasal cavity is a major reservoir for pathogens that can spread from there to other sections of the respiratory tract and become involved in diseases such as asthma, allergic rhinitis, chronic rhinosinusitis (CRS) or pneumonia, among others. A healthy nasal microbiome is characterized by highly regulated microbial interactions, where a variety of immune and structural cells produce different biomarkers and reflect biological events. Some examples are monokine induced by interferon γ (MIG), IP-10, monocyte chemoattractant protein (MCP-1), eotaxin, and epidermal growth factor (EGF), in addition to several interleukins, such as IL-15, IL-8, IL-1α and IL-1β, involved in modification of proinflammatory responses [117].

An example of application of electrochemical biosensing to the analysis of nasal fluid is the method developed by Hassan-Nixon et al. [118] involving a label-free impedimetric immunosensor for the determination of the IL-8 present in the nasal epithelial lining fluid (NELF). Polyclonal anti-IL-8 antibodies were immobilized on a gold electrode modified with cysteamine and the anti-fouling zwitteronic hydrogel polycarboxybetaine methacrylate (pCBMA) prepared by photopolymerization in the presence of ethyleneglycol dimethylacrylate (EGDMA). Impedimetric responses using Fe[(CN)_6_]^3−/4−^ as redox probe provided a logarithmic calibration with a linear range between 500 fg mL^−1^ and 50 ng mL^−1^ and a LOD value of 90 fg mL^−1^. The high sensitivity of the developed immunosensor was attributed to a superior binding affinity of the antibody due to the stabilizing effect of the ammonium and acetate ions present in the polycarboxybetaine moiety of pCBMA and to the super-hydrophilicity of the polymer, which resulted in the removal of water molecules from the hydrophobic regions of the antibody thereby increasing the protein–substrate binding affinity. The NELF samples, collected by means of a nasosorption device [119] gently introduced into the nostril lumen, were directly analyzed without the need for treatment obtaining satisfactory results.

Metabolomics has provided novel insights into the biomarkers and mechanisms of CRS [120], a chronic disease characterized by sinonasal mucosal inflammation in which commensal microbes, pathogens and their products play leading roles [121]. Two types of CRS can be distinguished: eosinophilic CRS with polyps, where interleukins IL-5 and IL-33 are mainly associated to inflammation, and non-eosinophilic CRS (without polyps), characterized by the presence of interferon gamma (IFN-γ), and the interleukins IL-17A, IL-1β and IL-8. Moreover, some of these, in particular those related to inflammation, may also be found in nasal lavages from patients of obstructive sleep apnea (OSA) [122].

A large body of epidemiologic evidence has been published linking OSA with important cardiovascular conditions, including hypertension, metabolic syndrome, coronary artery disease, arrhythmia, and heart failure [123]. Given the potentially serious consequences of untreated severe OSA, timely recognition, risk stratification, and appropriate treatment are crucial. It is well known that OSA may lead to an inflammatory response, and significant OSA is characterized by a distinct biomarker profile including significantly higher IL-6 levels after sleep in patients with moderate/severe OSA in comparison with individuals with mild or no disease [124]. Monitoring of IL-6 secretion in harvested cells and in vivo with a voltammetric immunosensor was reported [125]. A label-free configuration involving immobilization of anti-IL-6 capture antibodies on gold nanowires modified with graphene oxide (GO) and 4-aminophenyl phosphorylcholine to minimize nonspecific adsorption was implemented, and a sandwich-type assay was established by means of anti-IL-6 detection antibodies conjugated to GO and integrated with Nile blue as the redox probe. The electrochemical responses using SWV provided a linear range of 1–300 pg mL^−1^ with the lowest detectable concentration at 1 pg mL^−1^. The resulting method was applied to cell and in vivo analysis by monitoring IL-6 secretion in mouse brain.

Sinus mucosal cells also produce a large number of proteins and peptides with antimicrobial functions, including enzymes (e.g., lysozymes), defensins such as human beta-defensin-2 (hBD-2), and members of palate lung and nasal epithelium clone (PLUNC) family, whose levels are decreased in patients with nasal polyps, subsequently affecting the microbial colonization of the nose and sinuses in these individuals. Nasal epithelial cells were shown to express significantly higher levels of the pro-remodeling factors vascular endothelial growth factor (VEGF) and TGF-β cytokine compared to healthy individuals. Furthermore, local expression of the chemokines CCL-11 (eotaxin-1) and CCL-26 (eotaxin-3) is increased in asthma and allergic rhinitis. In these patients, nasal mucosa shows seasonal changes, including increased neutrophil levels expressing integrin proteins such as CD11b and glycoproteins such as CD66b and CD63.

A variety of electrochemical biosensors have been developed for the determination of VEGF. Among them, the multiplexed configuration prepared by Shen et al. [126] for the simultaneous quantification of VEGF, TGF-β and IFN-γ can be highlighted. As Figure 8 shows, a gold electrode modified with graphene oxide and streptavidin was used for the immobilization of the biotinylated aptamers respectively conjugated with anthraquinone (AQ), ferrocene (Fc), and methylene blue (MB). Binding of specific targets induced unfolding of aptamer hairpin structure, leaving the redox markers far from the electrode and reducing the electron-transfer efficiency. Thus, the redox peak currents from the electroactive labels decreased with increasing target levels in the 5–300 pg mL^−1^ VEGF, 5–200 pg mL^−1^ TGF-β, and 5–300 pg mL^−1^ IFN-γ ranges. This multiplexed aptasensor was applied to the analysis of sweat and serum samples. The analytical characteristics of other methods for these analytes involving electrochemical biosensors are summarized in Appendix A [127,128].

## 5. Concluding Remarks and Future Perspectives

In this review article, recent applications of electrochemical (bio)sensors for monitoring microbial metabolites and detecting biomarkers of diseases related to different types of human microbiome are discussed. (Bio)sensing devices have been reported for important metabolites such as trimethylamine N-oxide (TMAO), short-chain fatty acids (SFCAs), myeloperoxidase (MPO), alkaline phosphatase (ALP), metalloproteinases such as MMP-9, and cytokines such as IL-8 or VEFG (vascular endothelial growth factor), among others. The developed analytical methodologies show, in general, excellent sensitivity and rapidity of implementation, and are competitive with other commonly used methods, mainly ELISA. In addition, a remarkable advantage of electrochemical biosensors is their ability to be employed in multiplexed analyses, as has been demonstrated in cases such as the simultaneous determination of autoantibody biomarkers of rheumatoid arthritis (RF and anti-CCP) or cytokines related to inflammation (IL-1β and TNF-α). Except for a few sensors, applied to the detection of electroactive biomolecules, such as TMAO, SCFAs, indole, and 5-hydroxyindole-3-acetic acid (5-HIAA), practically all the electroanalytical methods developed are based on biosensors involving antibodies or, to a lesser extent, DNA strands or aptamers. Regarding the electrochemical techniques, the significant increase in applications of electrochemical impedance spectroscopy is noteworthy. The use of this technique makes it possible to prepare biodetection platforms with no need for enzyme labels, thus saving time and reagents. Special attention is deserved by the use of nanomaterials such as gold nanoparticles, carbon nanotubes and graphene oxide, which allow the preparation of biosensors with improved immobilization strategies and/or electron transfer rates with the electrode surface.

However, it is essential to highlight the need for future advances in the application of the developed methods to real samples. This makes it necessary to establish biosensing strategies and transduction schemes that are not only highly sensitive and selective, but also robust enough to be applied to a wide variety of samples in which biomarkers of microbial metabolism and others related can be found. It should be taken into account that such biological samples (feces, nasal fluids, tissues, mucosal and gingival fluid, among others) are much more complex and diverse than human serum or plasma, samples where electrochemical biosensors are usually validated. Therefore, adequate procedures and means to collect representative samples are also required, as well as to promote the fabrication of portable and implantable bioelectronic devices for biomarker detection, especially in the fields of oral and nasal microbiomes. Although the interest in saliva monitoring has increased enormously in recent years, relatively few studies have been focused on developing in-mouth biosensing platforms for salivary proteins related to the oral microbiome. Once the diagnostic power of saliva has been demonstrated, advances in the preparation of biocompatible materials and the use of non-fouling materials/surfaces will presumably allow rapid progress in this area of research.

The tremendous advances that electrochemical biosensors have experienced and demonstrated in recent years, as can be deduced from the examples discussed in this review article and from the recent literature, have mainly benefited the fields of oncology and immune diseases. This allows predicting that this type of biosensor will be similarly successful in contributing to advances in diseases and/or disorders related to the human microbiome. It is a matter of transferring all that has been learned and demonstrated in the field of cancer and immune diseases to the no less fascinating and complex world of the human microbiome.

## Figures and Tables

**Figure 1 sensors-23-00837-f001:**
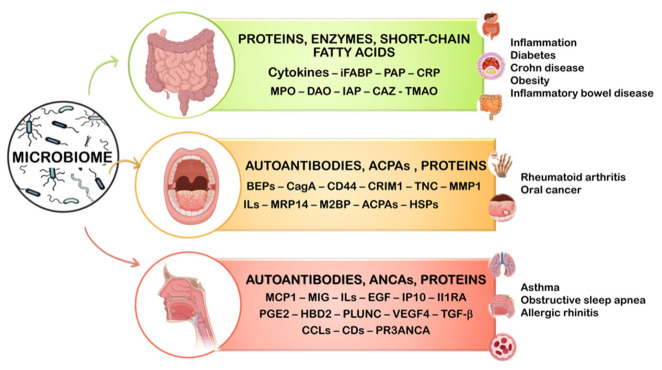
Selected gastrointestinal, oral, and nasal microbiome biomarkers and related diseases. Figure prepared by the authors.

**Figure 2 sensors-23-00837-f002:**
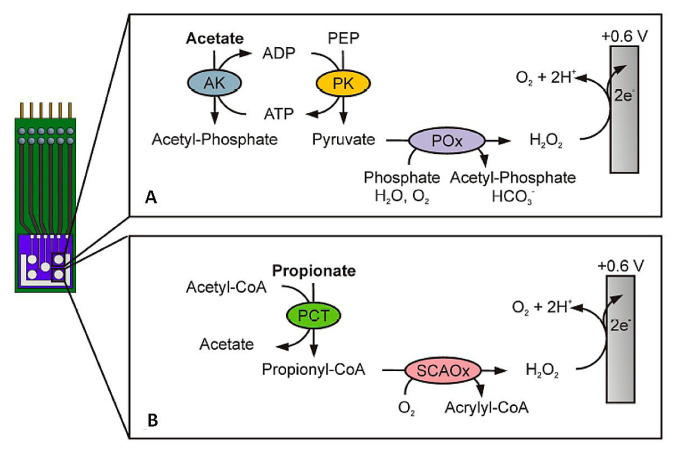
Amperometric detection principles of (**A**) an acetate biosensor using acetate kinase (AK), pyruvate kinase (PK), and pyruvate oxidase (POx), and (**B**) a propionate biosensor using propionate CoA-transferase (PCT) and short-chain acyl-CoA oxidase (SCAOx). Adapted from [34] with permission.

**Figure 3 sensors-23-00837-f003:**
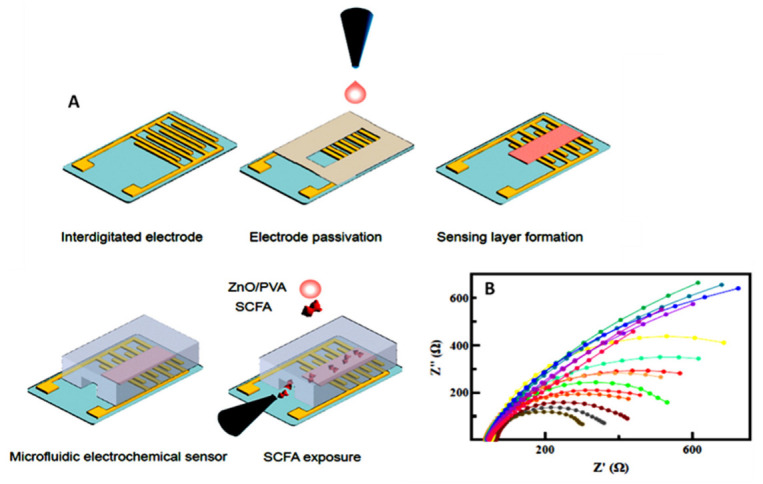
(**A**) Schematic view of the microfluidic setup and sensing layer synthesis for the determination of SCFAs and (**B**) Nyquist spectra of the sensing layer at various concentrations of acetic acid and propionic acid in a mixture with 0.5 mg mL^−1^ butyric acid. Adapted from [35] with permission.

**Figure 4 sensors-23-00837-f004:**
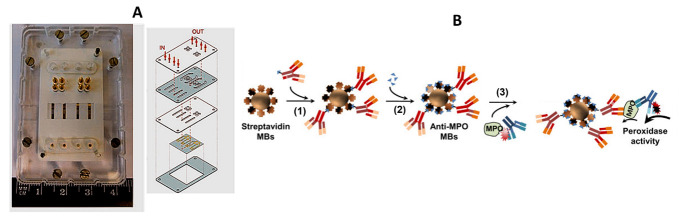
(**A**) Microfluidic chip and holder, and the different components of the cartridge. (**B**) Scheme of the magneto-immunoassay format for the determination of MPO: (1) anti-MPO-biotin; (2) biotin used as blocker; (3) anti-MPO-HRP. Adapted from [59] with permission.

**Figure 5 sensors-23-00837-f005:**
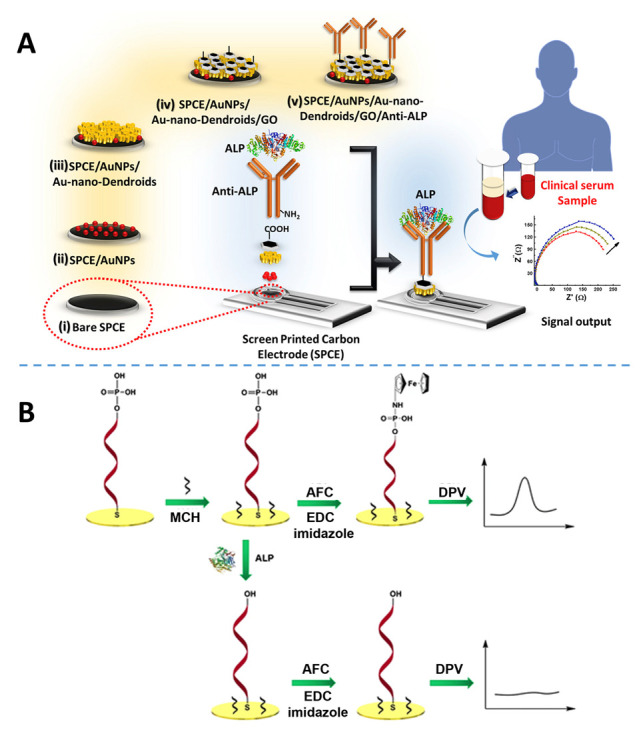
Schematic illustrations of: (**A**) the SPCE/AuNPs/Au-nano-dendroids/GO/anti-ALP probe for the determination of ALP in serum, and (**B**) the electrochemical assay of ALP activity based on the enzyme-catalyzed reaction. Adapted from [66,67], respectively, with permission.

**Figure 6 sensors-23-00837-f006:**
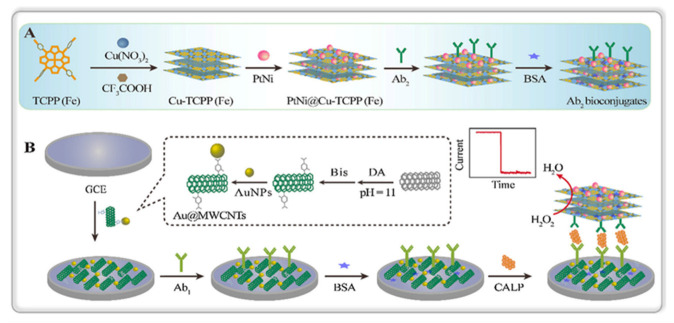
Illustrative schemes of: (**A**) the preparation procedure of PtNi@Cu-TCPP(Fe)-Ab_2_ bioconjugate, and (**B**) the construction of the sandwich immunosensor. Reprinted from [74] with permission.

**Figure 7 sensors-23-00837-f007:**
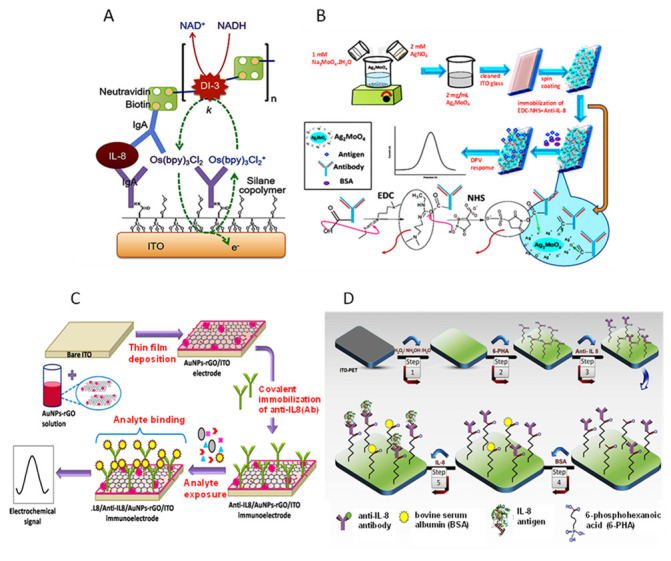
Schematic illustrations of some electrochemical immunosensors for the determination of IL-8: (**A**) a sandwich-type immunoassay using a polyenzyme label on diaphorase (DI-3) and neutravidin; (**B**) the synthesis of β-Ag_2_-MoO_4_ NPs and immunoelectrode fabrication; (**C**) fabrication of an AuNPs-rGO based immunosensor; (**D**) steps for preparation of an impedimetric immunosensor. Reprinted from (**A**) [101], (**B**) [102]; (**C**) [103] and (**D**) [104] with permission.

**Figure 8 sensors-23-00837-f008:**
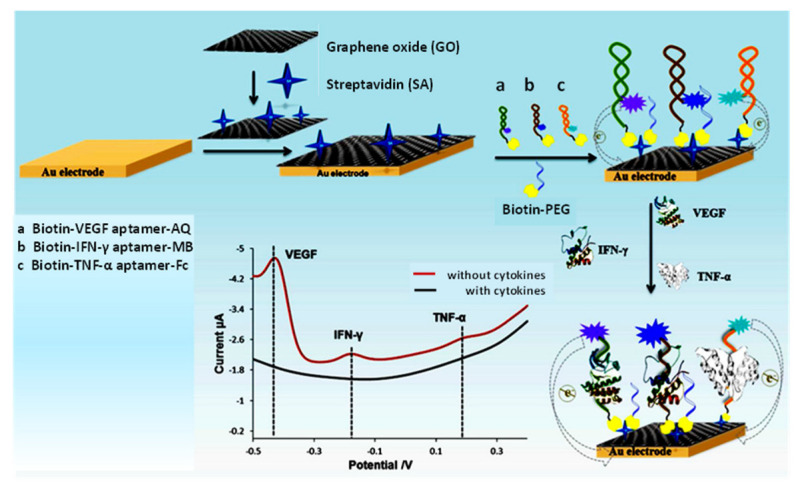
Schematic of an electrochemical aptasensor for the simultaneous and real-time monitoring of VEGF, IFN-γ and TNF-α. Reproduced from [126] with permission.

## Data Availability

Not applicable.

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
