# Peer review of "Electrochemical (Bio)Sensing Devices for Human-Microbiome-Related Biomarkers"

_sensors, 2023, doi:10.3390/s23020837_

Round 1

Reviewer 1 Report

After reviewing this manuscript, it seems to be an interesting article. However, there are some points to be considered before recommending it for publication.

1- The abstract section is recommended to be rewritten to be more explanatory.

2- Some complex sentences arise through the whole manuscript. Shorter and more comprehensive ones will be better.

3- The authors have to point out to the novelty (scientific contribution) of this review when compared to published ones.

4- The quality of Figure 3 should be improved.

5- The legends of Figure 8 are not legible.

6- Some relevant references published in 2023 can enrich this article.

Author Response

After reviewing this manuscript, it seems to be an interesting article. However, there are some points to be considered before recommending it for publication.

We really would like to thank this reviewer very much for his/her work and the interesting issues pointed out, addressed as indicated below.

1- The abstract section is recommended to be rewritten to be more explanatory.

Thank you for this comment. We have rewritten the abstract as follows to make it more explanatory:

The study of the human microbiome is a multidisciplinary area ranging from the field of technology to that of personalized medicine. The possibility of using microbiota biomarkers to improve the diagnosis and monitoring of diseases (e.g. cancer), health conditions (e.g. obesity) or relevant processes (e.g. aging) has raised great expectations, also in the field of bioelectroanalytical chemistry. The well-known advantages of electrochemical biosensors: high sensitivity, fast response and possibility of miniaturization, together with the potential of new nanomaterials to improve their design and performance, position them as unique tools to provide a better understanding of the entities of the human microbiome and raise the prospect of huge and important developments in the coming years. This review article compiles recent applications of electrochemical (bio)sensors for monitoring microbial metabolites and disease biomarkers related to different types of the human microbiome, with a special focus on the gastrointestinal microbiome. Examples of electrochemical devices applied to real samples are critically discussed, as well as the challenges to be faced and where future developments are expected to go.

2- Some complex sentences arise through the whole manuscript. Shorter and more comprehensive ones will be better.

We appreciate this comment. We have carefully revised the entire text by shortening and simplifying some sentences.

3- The authors have to point out to the novelty (scientific contribution) of this review when compared to published ones.

Thank you very much for this comment. We have clarified this important aspect in the revised manuscript (end of section 1) as follows:

…However, to our knowledge there is not reported any review article where the opportunities and potential of electrochemical biosensors to advance the knowledge and the analysis of the human microbiome are critically discussed. That is why this review article aims to provide an overview of recent applications of electrochemical (bio)sensors for the monitoring of microbial metabolites and the detection of disease biomarkers related to different types of the human microbiome, with a special focus on the gastrointestinal microbiome. To illustrate the critical discussion, examples of electrochemical devices applied to real samples are considered…

4- The quality of Figure 3 should be improved.

We have replaced Figure 3 with another Figure with higher quality.

5- The legends of Figure 8 are not legible.

Figure 8 is reprinted from reference 124 (new ref. 126) and we cannot modify the legend, so we have tried to improve the resolution of the Figure and have enlarged it to make the legend more legible.

6- Some relevant references published in 2023 can enrich this article.

Thank you very much for this comment. We have only found one reference in press probably published in 2023 that we have included in the revised version (new reference 4).

Reviewer 2 Report

sensors

Review 1

Electrochemical (bio)sensing devices for human microbiome related biomarkers

Authors have made a attempt on reviewing the recent applications of electrochemical (bio)sensors 

for monitoring microbial metabolites and detecting biomarkers of diseases related to different types

of the human microbiome.

I wish to see atleast a table comparing as this was review, it must be there, for comparison of several parameters.

Authors should include that.

If the table is there, there can be more discussionsor exchange of ionions/soem insight into it will follow.

Separate Conclusions and outlook, in the manuscript it was as summary and conlcuions/future perspectives etc.

Figures must be with more clarity

References may not be sufficient

Accept for publication with minor mandatory changes

Author Response

Electrochemical (bio)sensing devices for human microbiome related biomarkers

 Authors have made a attempt on reviewing the recent applications of electrochemical (bio)sensors for monitoring microbial metabolites and detecting biomarkers of diseases related to different types of the human microbiome.

We are also very grateful to this reviewer for his/her work and comments, considered as detailed below.

I wish to see at least a table comparing as this was review, it must be there, for comparison of several parameters.

Authors should include that.

If the table is there, there can be more discussions or exchange of ionions/soem insight into it will follow.

We fully agree with the reviewer on the necessity and usefulness of a comparative table. Therefore, already in the initial version manuscript, we provided in the Supplementary Material a very extensive and detailed table (Table S1) comparing the most representative characteristics of 37 selected electrochemical (bio)sensors, grouping them according to the type of microbiome to which they have been applied.

Separate Conclusions and outlook, in the manuscript it was as summary and conlcuions/future perspectives etc.

Since separating them would result in very short sections and it is common to discuss these aspects in the same section in review articles, we prefer to leave them together and have changed the title of section 5 to “Concluding remarks and future perspectives”.

Figures must be with more clarity

We have revised the Figures to provide them with the highest possible resolution.

References may not be sufficient

We appreciate this comment. Although we consider that 126 references is an acceptable and sufficient number for a review article on a recent topic, we have included two additional references (new references 4 and 10).

These References were inserted in the new text as follows:

“…and treatment strategies [3]. In a very recent review article, Aggarwal et al. [4] describe the current understanding of the relationship between microbiome and diseases as well as the therapeutic effects of microbiome modulation on the host. Microbial cells are…”

“…the “fifth organ” [7]. The most recent studies on the involvement of gut microbiota in the pathogenesis of many diseases as well as the different strategies used to manipulate the gut microbiota in the prevention and treatment of disorders have been reviewed [10]. The outcomes of this…”

Accept for publication with minor mandatory changes

Thank you very much for recommending the publication of our work after addressing these minor changes pointed out.

Reviewer 3 Report

the review article is sound very attractive to a broad audience but it needs some modifications as the following:

1- I think that figure 1 needs some modification, especially for panel B, not clear (preferable to modulate it to be easily readable)

2- It will be better if you draw a diagram or illustrative figure of the gut microbiome and/or its biomarkers to explain its role in the regulation of metabolism, digestion, and disease-causing, cancer,.... 

3- focus on the importance of the electrochemical biosensor and their sensitivity in disease and disease-causing diagnosis

4- subdivide the gastro-microbiome biosensors into subdivisions (i.e. proteins, peptides (cytokines), enzymes....etc. and make the figures collective for that in one figure

follow the same in the oral and nasal

5- please modify the figures with high resolutions

Author Response

the review article is sound very attractive to a broad audience but it needs some modifications as the following:

Thank you very much for your work and kind comments.

1- I think that figure 1 needs some modification, especially for panel B, not clear (preferable to modulate it to be easily readable)

2- It will be better if you draw a diagram or illustrative figure of the gut microbiome and/or its biomarkers to explain its role in the regulation of metabolism, digestion, and disease-causing, cancer,.... 

Thank you very much for your suggestions. The figure proposed by the Reviewer would be very appropriate. However, these types of figures already illustrate other articles related to specific topics included in this review. Therefore, with the aim of differentiating this more general review from other previous articles and at the same time following the Reviewer's recommendation, we have replaced Figure 1 for another illustration.

3- focus on the importance of the electrochemical biosensor and their sensitivity in disease and disease-causing diagnosis

Thank you again for this suggestion. Indeed, we fully agree that the sensitivity of biosensors is crucial in the diagnosis of diseases and their causes. For this reason, data on the detection limits and linearity intervals achieved with the selected configurations are provided throughout the manuscript and in Table S1.

4- subdivide the gastro-microbiome biosensors into subdivisions (i.e. proteins, peptides (cytokines), enzymes....etc. and make the figures collective for that in one figure

follow the same in the oral and nasal

Once again, we are very grateful for this suggestion. However, we do not agree with the division proposed by the reviewer, as the types of biomarkers considered in each microbiome often have overlapping properties. We also do not understand the need to separate proteins from peptides or cytokines or enzymes since they are all protein biomarkers and their activity is closely related.

5- please modify the figures with high resolutions

Thanks again. Following your suggestion and those of the other Reviewers we have improved the figures.

Round 2

Reviewer 1 Report

The revised version of this manuscript appears in a more qualified form more than the previous one. 

The reqiured queries are answered conveniently. I can recommend it for publication in the present form.

Reviewer 3 Report

accept in the present form